



# Historic photographs of glaciers and glacial landforms from the R. S. Tarr collection at Cornell University

Julie Elliott[1], Matthew E. Pritchard[2]

[1]Department of Earth, Atmospheric, and Planetary Sciences, Purdue University, West Lafayette, IN 47907
[2]Department of Earth & Atmospheric Sciences, Cornell University, Ithaca, NY 14850

*Correspondence to*: Julie Elliott (julieelliott@purdue.edu)

**Abstract.**

Historic photographs are useful for documenting glacier, environmental, and landscape change and we have digitized a collection of about 1949 images collected during an 1896 expedition to Greenland and trips to Alaska in 1905, 1906, 1909,
and 1911, led by Ralph Stockman Tarr and his students at Cornell University. These images are openly available in the public domain through Cornell University Library (http://digital.library.cornell.edu/collections/tarr; *Tarr and Cornell University Library* (2014), http://doi.org/10.7298/X4M61H5R). The primary research targets of these expeditions were glaciers (there are about 990 photographs of at least 58 named glaciers) but there are also photographs of people, villages, and geomorphological features, including glacial features in the formally glaciated regions of New York state. Some of the
glaciers featured in the photographs have retreated significantly in the last century or even completely vanished. The images document terminus positions and ice elevations for many of the glaciers and some glaciers have photographs from multiple viewpoints that may be suitable for ice volume estimation through photogrammetric methods. While some of these photographs have been used in publications in the early 20[th] century, most of the images are only now widely available for the first time. The digitized collection also includes about 300 lantern slides made from the expedition photographs and
other related images and used in classes and public presentations for decades. The images are of scientific interest for understanding glacier and ecological change, of public policy interest for documenting climate change, of historic and anthropological interest as local people, settlements, and gold-rush era paraphernalia are featured in the images, and of technological interest as the photographic techniques used were cutting-edge for their time.

## 1 Background

In recent decades, glacier retreat has become symbolic of climate change, but the relationship between climate and glacier response is complex. While global trends indicate significant ice loss throughout the 20[th] and early 21[st] centuries, glaciers are not losing ice at the same rate and a small fraction are continuing to gain mass (e.g. *Larsen et al.*, 2007; *Larsen et al.*, 2015; *Hill et* al, 2017). A variety of factors control whether an individual glacier is advancing or retreating and how it will respond to regional climate change (e.g. *Post et al.*, 2011; *Larsen et al.*, 2015). To better understand the link between



climate and glacier behaviour, long-term records beyond the relatively short temporal limits of satellite observations are essential. Historic photographs can significantly expand the time span of observations, leading to both qualitative and quantitative evaluations of glacier fluctuations and their possible causes (e.g. *Molnia*, 2007; *Bjork et al.*, 2012).

Here we describe a newly digitized collection of photographs from a series of expeditions undertaken in the late 19[th] and
early 20[th] centuries to study glaciers and other geographical features in Greenland and Alaska led by Professor Ralph Stockman Tarr and his students from Cornell University. Tarr was a professor of physical geography with particular interests in glaciology and geomorphology. In pursuit of these interests, he led or participated in expeditions to western Greenland in 1896 and to various regions of Alaska and western Canada in 1905, 1906, 1909, and 1911. His former student and frequent collaborator, Lawrence Martin, joined Tarr on the expeditions of 1905, 1909, and 1911 and made trips to
Alaska without Tarr in 1904, 1910, and 1913 (*Tarr and Martin*, 1913). The collection presented here includes images from the 1905, 1906, 1909, and 1911 expeditions and the 1896 Greenland expedition. The expeditions are discussed further below. In addition to the images from expeditions, the collection includes digitized images of lantern slides (glass slides used in magic lanterns, an early version of a projector) that were used in teaching at Cornell and public lectures. These lantern slides duplicate a few of the original images from the expeditions but also include images from other scientific expeditions
as well from commercial teaching collections. A summary of the full collection is shown in Table 1. Approximate image locations are shown in Figs. 1 and 2 (Alaska) and Fig. 3 (Greenland).

Counting the expedition images and lantern slides together, there are images of at least 49 named glaciers in Alaska and eight in Greenland in the collection (Tables 2 and 3). These glaciers featured in these images are of global importance as
glaciers in coastal Greenland and Alaska are significant contributors to current sea-level rise because of their rapid loss of ice mass (e.g., *Gardner et al.*, 2013). Of these 49 glaciers, 34 have photos that clearly show the majority of their terminus, which will allow the position to be mapped and compared to modern terminus positions. Roughly half of the glaciers have images in which the vertical extent of ice is easily distinguished against valley walls and other features that can serve as benchmarks for modern comparisons. Eight of the glaciers in Alaska have photographs of the terminus region taken from at
least three different viewpoints, which may make the images suitable for ice volume estimation through photogrammetric methods. About 20% of the glaciers, either through single photographs or a combination of photographs, have imagery covering at least 5 km of their length, measured from their terminus. Tables 2 and 3 list which glaciers fall into each of these categories.

In addition to the glaciers themselves, Tarr was very interested in the landforms formed and left behind by the glaciers. The collection includes images of alluvial fans, various types of moraines, outwash plains, eskers, kettles, fosse, and other glacial features (Fig. 4). In Alaska and Greenland, the images show active and recently active features. The collection also includes images of features developed during the last glacial maximum in the Finger Lakes region of upstate New York. Moraines

are the most frequently featured glacier landforms in the images. In many images, the moraine appears alongside other

features, such as mountains or shorelines,that are easily located in modern maps and imagery. This is especially true in Alaska, where moraine locations in images from Prince William Sound (Columbia, Spencer, and Shoup glaciers), the Wrangell Mountains (Kennicott Glacier), and the Yakutat Bay/Russell Fjord region (Hubbard, Orange, Hidden, Variegated, Atrevida, and Hidden Glaciers) can be mapped and compared to present day landscapes. Other types of geological changes are also documented in these photographs.  One goal of the expeditions to Alaska was to document changes caused by a

series of earthquakes in the area (e.g., *Tarr*, 1909; *Martin*, 1910; *Tarr and Martin*, 1912a) that caused significant, abrupt uplift and subsidence (Fig. 5).  These observations have been used in modern tectonic studies (*Plafker and Thatcher*, 2008) and can be useful in separating instantaneous tectonic motion from the effects of glacial isostatic adjustment that have accumulated over the past century.  For all of the regions Tarr visited, the collection includes images of people, towns, and smaller settlements that provide a glimpse into life at the turn of the 20[th] century. particularly gold rush era interior Alaska,

the Yukon, and British Columbia.

Historic photographs have been used for decades to observe glacier change (e.g., *Molnia*, 2007; 2008; *Meier et al*., 1985), but the digitized photographs described here are a significant addition as they have been little studied and include glaciers with few historic photographs.  Although most of the photographs in this collection have been publically available in the

Division of Rare and Manuscript Collections (RMC) of the Cornell University Library for decades, only a fraction have been published in articles (e.g., *Tarr and Martin*, 1914).  In particular, photographs from the 1911 expedition were not used extensively in publications (although see, e.g., *Tarr and Martin*, 1912b; 1913; *Martin* 1913a), because Tarr died suddenly in March, 1912 at age 48  and the collaborators moved on to other projects (e.g., *Brice*, 1985; 1989).  Thus, many of the photographs have not been otherwise published or catalogued and have been seldom viewed over the past 100 years.


In the following sections, we describe the purposes and context of the expeditions, the types of photographs and the subjects, and how they were digitized.

## 2. The Expeditions and Photography

### 2.1  Photographers and Equipment

The photographs were taken by a variety of photographers.  Tarr and Martin both took photographs.  During the 1906 and 1909 expeditions, many of the photos were taken by Oscar D. von Engeln.  A keen photographer, von Engeln worked with Tarr as an undergraduate and graduate student at Cornell and later become a professor in the Department of Geology and Geography there.  James Otis Martin, a Cornell student who accompanied Tarr to Greenland in 1896, took a number of the photographs during that expedition (*Tarr,* 1897a; *Tarr,* 1897i).  Photos in the collection were also taken by members of the



U.S. Geological Survey, engineers of the Copper River and Northwestern Railway, members of the Canadian Boundary Survey, and several unnamed Alaskan photographers (*Tarr and Martin*, 1914). The lantern slides include photos taken by members of other well-known expeditions, including William Libbey, a Princeton geographer who participated in a trip to explore Mount St. Elias in Alaska in 1886, Peary's 1894 expedition to Greenland, and a 1899 Princeton-funded trip to Greenland (*Koelsch*, 2016); F. Jay Haynes, a professional photographer who visited Alaska, Yellowstone, and other parts of

the American West; Henry Fielding Reid, a professor at Johns Hopkins who performed pioneering studies of glacier dynamics in southeast Alaska in addition to ground breaking work on how faults related to earthquakes; and Israel Russell, a USGS scientist who explored the regions of Mount St. Elias and Yakutat Bay in Alaska.

Equipment varied depending on the trip and the most detail is known about the equipment used in the 1906 and 1909 Alaska expeditions as von Engeln published articles on techniques he used to take and develop photographs in the challenging field

conditions (*von Engeln,* 1907b; 1910). During those expeditions, he used a Rochester Optical Company Pony Premo self-casing folding plate camera (which is preserved in the Cornell RMC, see Fig. 6a,b) as well as several other cameras including multiple Kodak film cameras (*von Engeln,* 1907b). Exposures were made on glass plates and film negatives (typically standard 4 x 5 inch size, with a few of larger size. Several lenses were used, including a long-focus lens that was custom made by Bauch and Lomb for the Alaska work (*von Engeln,* 1907b). At least one camera had a mount system that

allowed the capture of panoramic images on film (Fig. 6e,f). As shown in Fig. 6c-e, some of the images capture the process and difficulty of taking photographs in the rugged field environment. The cameras, lenses, shutter, and plate holders were packed into a leather case with straps so that it could be more easily carried along with a tripod, plates and film, and containers for changing plates and protecting exposed film from the excessive humidity of southeast Alaska (*von Engeln,* 1907b). Although the camera system was designed to be compact, setting it up and making the exposures often took

considerably more time than scientific observations and notes at a site (*von Engeln,* 1910).

## 2.2 The Expeditions

We briefly describe the expeditions that are summarized in Table 1 and Figs. 1, 2, and 3.

2.2.1 **1896 Greenland**

     In 1896, Tarr, along with other faculty and students from Cornell travelled to Greenland as part of Robert Peary's expedition that attempted (and failed) to remove the largest of the three Cape York meteorites (*Tarr,* 1896; *Huntington*, 2002). The Cornell group was one of three scientific parties along on the expedition; after stops along the coast of Labrador and at Baffin Island, Disko Island, Vaigat Strait, and Umanak, they were landed on the Nugsuak Peninsula along the

Upernavik Archipelago (Fig. 3) where they stayed for several weeks making studies of the geology, plant life, birds, and invertebrates (*Tarr,* 1896). While there, they described and assigned names to a number of geographic features including



Cornell glacier, Wyckoff glacier (after an Ithaca businessman who provided financial backing for the expedition), and Mt. Schurman (after the President of Cornell University) (e.g., *Tarr,*, 1896; *Tarr* 1897a,b). Tarr wrote extensively of his field observations on the trip (*Tarr*, 1896; *Tarr* 1897a-i). The digitized photographs are from a variety of locations (but mostly

from the Nugsuak Peninsula) and show glaciers, geological features, local people, and some of the day-to-day activities and challenges of the expedition (Fig. 7).

### 2.2.2. **1905 Alaska**

In the summer of 1905, Tarr and Martin went to Yakutat Bay (Fig. 2a). Tarr was in charge of a USGS party

charged with a general geological survey of the region while Martin was funded by the American Geographical Society (e.g. *Tarr and Martin*, 1905). The scientific party made observations of surface changes the occurred after a series of M8 earthquakes that occurred in the region in 1899 (e.g. *Plafker and Thatcher*, 2008), general descriptions of glaciers in the Yakutat Bay area and evidence for the extent of past glaciation, and noted the return of vegetation to areas in which glaciers had recently retreated. The work done during this trip was the primary focus of *Tarr and Martin* (1905) *Tarr and Martin*

(1906a-c) and featured in a number of other publications (e.g. *Tarr* 1907a-e; *Tarr* 1909; *Tarr* 1910; *Tarr and Martin*, 1907; *Tarr and Martin*, 1912a; *Tarr and Martin*, 1914). Digitized photographs from this expedition include images of the area's glaciers, faults and other features related to the 1899 earthquakes, and glacial landforms (Fig. 8).

### 2.2.3. **1906 Alaska**

Tarr again led a USGS-sponsored party to Yakutat Bay in the summer of 1906. One of the party's objectives was to cross the Malaspina Glacier, but they discovered that the normally navigable tributary glaciers east of the Malaspina had advanced and created impassable crevasse fields (*Tarr*, 1907 d-e). At least two other glaciers in the Yaktuat Bay area had also advanced since the summer of 1905 (*Tarr and Martin*, 1912a). Scientific findings of the expedition are described in *Tarr* 1907a-e; 1909; *Tarr* 1910; *Tarr and Martin*, 1907; *Tarr and Martin*,1912a; *Tarr and Martin*, 1914; *von Engeln*, 1911..

Popular accounts of the expedition include *von Engeln* (1906, 1907a) and *Alley* (2012). In the collection , the 1906 digitized photos are primarily images of the glaciers along the eastern edge of the Malaspina Glacier and within Yakutat and Disenchantment Bays and Russell Fjord (Fig. 9).

### 2.2.4 **1909 Alaska**

With funding from the National Geographic Society, Tarr and Martin returned to Alaska with a scientific party including a dedicated photographer and topographer in 1909. Most of their time was spent in Yakutat Bay, but the group also travelled further west and made observations at Valdez, Columbia, and Shoup glaciers in eastern Prince William Sound, and Miles, Childs, and Allen glaciers along the lower Copper River (Fig. 1). The primary purposes of the trip were to make detailed maps of the glaciers of the southern Alaska coast and to make additional observations concerning the sudden

advance of glaciers noted in 1906 (Tarr and Martin, 1910a). During the trip, the party found that the glacier advances noted





in 1906 had slowed or stopped while two additional glaciers around Yakutat Bay showed signs of advance. Glaciers within Prince William Sound and the Copper River did not show evidence of significant change from earlier observations (e.g. *Tarr and Martin*, 1910a-b). Based on these results, Tarr and Martin proposed what they referred to as the earthquake advance theory or the glacial flood hypothesis: an earthquake causes an avalanche of snow and other material onto a glacier, causing

deformation that is transmitted down the glacier and eventually results in crevassing and an advance at the front of the glacier (e.g. *Tarr,* 1910; *Tarr and Martin*, 1910a). Later work in Alaska showed that the observed glacier surges are periodic and likely due to characteristics specific to the individual glacier system, not external factors such as earthquakes (e.g. *Post*, 1965; *Meier and Post*, 1969). Additional descriptions of the fieldwork and scientific work are found in Tarr and Martin (1912a; 1914) and *von Engeln* (1911). Within the collection, the 1909 expedition contributes more digitized images than any

other trip. The images show glaciers and glacial landforms in the Yakutat Bay, Prince William Sound, and Copper River regions as well as construction camps and waypoints along the route of the Copper River Railroad (Fig. 10).

### 2.2.5  1911 Alaska

The 1911 expedition, funded again by the National Geographic Society, was the most wide ranging of the Alaska

expeditions. While, Tarr, Martin, and the rest of the scientific party returned to a few previously visited sites around Prince William Sound, the overwhelming majority of locations had not been visited by Tarr or Martin. Members of the party spent time in Glacier Bay (Fig. 2), the Kenai Peninsula, and Prince William Sound before moving inland to the Wrangell Mountains. They then moved north into Interior Alaska, with stops at Fairbanks and other locations involved with gold mining. The group travelled up the Yukon River through Alaska and the Yukon (with a variety of stops including Dawson)

before reaching the headwaters of that river in British Columbia (Fig. 1). Scientific observations are presented in *Tarr and Martin* (1912b; 1913). The digitized images also have the most variety of any of the expeditions: glaciers in southeast and southcentral Alaska, railways, mining operations, roadhouses in Interior Alaska, city streets in Fairbanks and Dawson, and small settlements along the Yukon River (Fig. 11).

### 2.2.6  Ithaca and Upstate New York

The collection also includes images from closer to Tarr's home in Ithaca, New York. Over a forty year period (including 20 years at Cornell), Tarr accumulated images of glacial landforms and other geological features (including waterfalls) from Ithaca and upstate New York. Tarr used his observations in a number of publications on glacial erosion, the development of glacial landforms, and the geology of New York including *Tarr* (1904), *Tarr* (1905a-d), and *Tarr* (1906a-b).

Examples of digitized images from upstate New York are shown in Fig. 12.



## 3 Description of Dataset

### 3.1 Original Material

The original photographs are in the form of prints, glass plates, lantern slides, or negatives. The photographic material was placed by Tarr and his associates in individual paper envelopes with handwritten notes on the outside of the envelopes. As these envelopes were fragile due to age and not acid-free, the materials were re-housed in acid-free paper and stored with copies of the original envelopes. These materials are stored and publically accessible through the Cornell University Library Division of Rare and Manuscript Collections (RMC) as part of the Ralph Stockman Tarr Papers (collection number 14-15-92,

21 boxes) and the Oscar Diedrich von Engeln Papers (collection number 14-15-856, 18 boxes). Due to budget constraints, we could not digitize all of the images , but focused on images of glaciers and glaciated landscapes from the Alaska and Greenland expeditions. There are still hundreds of other photographs in the RMC that were not digitized along with thousands of lantern slides housed at the Department of Earth and Atmospheric Sciences at Cornell University.

### 3.2 Digitization

The photographs were digitized at the Cornell University Library. Transparencies (positives, negatives, film and glass) were scanned on Epson 10000xl flatbed scanners outfitted with transparency units using custom negative holders. Scan resolution varied depending on the size of the original (900 - 1800 dpi). The images were scanned to 16B, Grayscale, uncompressed

tiff files. Curve and level adjustments were made within the scan software at the time of scanning (epsonscan software, version 3.49A). After scanning there were additional curve adjustments made in Photoshop CS6 or CC2015 (version updated during project.) Master images were saved as 16B layered tiffs with curve adjustment layers. Uploaded access files are flattened, 8B, high quality compression, original sized jpegs. A few images were scanned with multiple exposures because the original image material were in poor shape and required more than one processing version to adequately show

details of different parts of the negative or glass plate. Several of the original film negatives and glass plates have been smudged or cracked over the years. In these cases, no corrective image processing has been attempted, so these artifacts appear in the digitized versions.

### 3.3 Data Availability


The digitized files (originally size, high quality jpegs) were uploaded into ITHAKA/Artstor's JSTOR Forum database and then made available through the Cornell University Library digital collections in the collection called "Historic Glacial



Images of Alaska and Greenland" (http://digital.library.cornell.edu/collections/tarr). The collection includes 1949 images with metadata, has a digital object identifier (http://doi.org/10.7298/X4M61H5R), and can be cited as Tarr and Cornell

University Library (2014)). Each image is linked back to the original file in JSTOR Forum which includes additional metadata (see below). The photographs are believed to have no known U.S. copyright or other restrictions. The Library does not charge for permission to use such material and does not grant or deny permission to publish or otherwise distribute public domain material in its collections. As a matter of good scholarly practice, we recommend that patrons using Library-provided reproductions cite the Library, the doi, and this article as the source of reproductions.


### 3.4  Metadata and Filenames

For each digitized image, the assigned filename follows a specific convention: a prefix ("tve" for Tarr and von Engeln), then

either the expedition year (e.g. "exp1911"), the phrase "lanternslide", or "Ithaca", and finally the sequential photograph number. For example, the 18th photo from the 1911 Alaska Expedition would have the filename "tve_exp1911_018". For images scanned with multiple exposures, the duplicates have filenames with numerical suffixes but otherwise identical information. Most of the photographs had descriptions (titles or captions) written on the original envelopes and these have been transcribed as the title of each digitized images. For the lantern slides, the captions were written on the slide itself

instead of the envelopes. In cases in which there was no title available, the image was given the title "No Label". For a few cases where we had other knowledge of the image subject, information on the likely location of the image was added. Besides the titles, a variety of other metadata is available for the images. For most photographs, the location or glacier name was included on the envelope or lanternslide. We were able to assign dates (either specific dates recorded on the envelopes or more general dates based on expedition diaries) to many of the images. The "work type" of the original image media is

also recorded (film negative, glass plate, or print). Both a general location (e.g. Alaska, Canada, Greenland) and a more specific location (e.g. Glacier Bay) are listed. A very few of the envelopes or lantern slides included the name of the photographer – for example, several of the Greenland photographs have "(Martin)" in the title referring to  J. O. Martin (discussed above) and some of the lantern slides taken from other collections have the photographer's name in the title (e.g., Reid, Haynes, Libbey). This information is retained in the title in the metadata when available.


As mentioned above, the JSTOR Forum database includes more fields of metadata than the Cornell Library database. These include the "Repository" or the box number and RMC Collection where the original image can be found as well as the "Accession Number" which is a number (specific to each expedition) written on the envelopes for the photographs – note that we did not use these numbers in naming the digitized files because not all photos had numbers and many were out of

order or missing. The lantern slides had a systematic naming convention (e.g., most images fall into the "North America" or "Glacier" categories) and numbering system that was indexed in a hand-written ledger in the RMC that was used to find the





slides for teaching and public presentations. For these lantern slides, the "Accession Number" in JSTOR Forum refers to this number (which is sometimes visible on the digitized slide) and the "Subject" gives the category used in the systematic naming convention. In some cases, the envelope includes some "additional notes" that have not been included in the digital

published metadata through the Cornell Library, but that are included in Table S1. For example, some images included a letter grade for the quality of the image (A+ being the highest and D being the lowest), presumably assessed by Tarr or von Engeln. In some other images, these notes include the name of the photographer, or if the photograph is duplicated as a lantern slide set, the number of the lantern slide is given (see above). Table S1 also includes the appropriate USGS Quadrangle topographic map or Natural Resources Canada topographic map for the images when available as well as a

direct link to each image.

We have not edited the glacier names in the metadata – it is possible that some photos labelled to show a certain glacier do not actually include that glacier and it is also possible that photos that do not include the name of glacier could include one. In several instances, names of glaciers used by Tarr and his colleagues were either not official names or were names that

were subsequently changed. In these cases, the glacier is indexed by the name used by Tarr and the currently accepted name is listed in the tables and the supplemental metadata. For Greenland glaciers, our choice of accepted name was guided by *Bjork et al*. (2015). Each image was assigned a general region (designated a "region" in the metadata - e.g. Canada) and a more specific area (designated a "sub-region" in the metadata - e.g. Yukon) when possible (for some images, location information was too vague to assign a more specific area). Several issues arose while assigning place names. One

concerned the fact that multiple locations (sometimes separated by significant distances) shared the same name. As an example, there were Serpentine Glaciers in Prince William Sound and Yakutat Bay in Alaska. In Greenland, there are multiple Nugsuak Peninsulas and Devil's Thumbs. To make accurate location designations, we relied on locations of images taken within the same time frame, publications discussing the images and expeditions, and, in a few cases, field diary entries. For these locations, we have added information to clarify which place is being referred to in the image (e.g. Nuussuaq

Peninsula (Upernavik Archipelago). Another issue concerned spelling of place names. Image subjects and locations recorded by Tarr and his colleagues had variable spellings for the same place (e.g. Nugsauk, Nugssauk) that in same cases differ from the current commonly accepted spelling (e.g. Nuussauq). We did not standardize or otherwise change the spellings of the transcribed titles. For instances where a designated sub-region tag contained a variably spelled name, we used the current common spelling. Image titles also contained variable spellings of other words (e.g. fjord/fiord,

canyon/canon); we left the spelling as it was transcribed.

4. **Complimentary Collections**

The images discussed in this paper represent a unique addition to publically accessible imagery of glaciers and glacier landforms in Alaska and Greenland that are not available elsewhere, but there are other collections that provide

complimentary imagery and information. Along with the expeditions discussed above, Lawrence Martin went on



expeditions to Alaska without Tarr in 1904 (e.g., *Tarr and Martin*, 1906), 1910 (*Martin*, 1911; 1913b), and 1913 (*Martin*, 1913c). Photographs and field notebooks from these trips are part of the William O. Field Papers in the Alaska and Polar Regions Collections at the University of Alaska, Fairbanks. Twenty-four photographs from Martin's trips are part of the ~15,000+ images the National Snow and Ice Data Center (NSIDC)'s Glacier Photograph Collection

(https://nsidc.org/data/glacier_photo/index.html). The NSIDC also holds and has scanned Martin's field notebooks from the 1904, 1905, and 1906 Alaska expeditions. Tarr's field notebooks from the 1896 Greenland expedition are in the RMC at Cornell. The RMC also includes a hard copy of the 1896 Greenland trip diary published by *Hoppin* (1897) that includes Tarr's hand-written notes in the margins. Tarr's field notebook from the 1905 Alaska expedition and his field notebook and personal diary from the 1906 Alaska trip are housed at the RMC. Von Engeln's 1906 trip diary and letters to his parents are

also in the RMC. Field notebooks from Tarr and von Engeln written during the 1909 Alaska expedition are at the RMC. None of these notebooks, letters, or diaries were digitized as part of this project. The NSIDC has scanned 19 pages of Tarr's field notebook from August 1909 as well as the second half of Tarr's field notebook from the 1911 Alaska expedition. In addition to the collection discussed in this paper, Cornell University previously digitized about 20 images related to Tarr and von Engeln which are available at:

https://digital.library.cornell.edu/?_=1532312108422&f%5Barchival_collection_tesim%5D%5B%5D=Oscar+Diedrich+von +Engeln+papers%2C+1896- 1964&f%5Bcollection_tesim%5D%5B%5D=Images+from+the+Rare+Book+and+Manuscript+Collections .

## 4 Conclusions

The newly digitized dataset will have a variety of uses for researchers. The images are of scientific interest for understanding glacier dynamics and ecological change, of public policy interest for documenting possible effects of climate change, and of historic and anthropological interest for capturing daily life in remote regions at the turn of the 20th century. The glacier images provide documentation of terminus positions and ice elevation and offer the possibility of ice volume estimates. Most the glaciers featured in the digitized images have undergone significant change over the past century and

comparison of the information in the images to modern data will provide new or more robust estimates of the extent of this change.

## Data availability

The digitized files are available through the Cornell University Library digital collections in the collection called "Historic

Glacial Images of Alaska and Greenland" (http://digital.library.cornell.edu/collections/tarr) and can be cited as.*Tarr and Cornell University Library* (2014), http:doi.org/10.7298/X4M61H5R.



**Author contributions**

J. E. conceived of this project, and both authors worked to collect the metadata, supervise undergraduate student workers,
secure funding, interact with Cornell Library staff to digitize the images and make them available online, and write the
article.

**Competing interests**

The authors declare that they have no conflict of interest.

**Acknowledgements**

Funds for digitization of the images were provided by Cornell University through the Grants Program for Digital Collections
in Arts and Sciences (Principal Investigator Aaron Sachs), through the Einhorn Discovery Grant and Undergraduate
Research Program of the College of Arts and Sciences (for student Emma Reed), and the Morley Research Fund from the
College of Agriculture and Life Sciences (for student Sam Nadell).  We thank all of the Cornell Library Staff who facilitated
the digitization, and made the data available on the web: Rhea Garen, Wendy Kozlowski, Jason Kovari, David Lurvey,
Hannah Marshall, Danielle Mericle, Liz Muller, and Melissa Wallace.  We thank the nine undergraduate students at Purdue
University and Cornell University who helped to catalog the Tarr and von Engeln collections and create the metadata:
Phoebe Dawkins, Anant Hariharan, Haydn Lenz, Alexis Lopez-Cepero, MacKenzie McAdams, Sam Nadell, Ella Noor,
Emma Reed, and Frank Tian.  Figures were generated with the Generic Mapping Tools software of *Wessel et al.* (2013).  We
are grateful to the late Art Bloom for making us aware of these photographs in the first place.





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



**Figure Captions**


Figure 1. Locations visited during expeditions to Alaska between 1905 and 1911.

Figure 2. Sites visited in southeast Alaska. (a) Sites visited in Yakutat Bay.. DB is Disenchantment Bay, RF is Russell Fjord, and NF is Nunatak Fjord. Numbered locations are: 1. Marvine Gl. 2. Blossom Island 3. Hayden Gl. 4. Floral Hills 5.

Floral Pass 6. Kwik Stream 7. Lucia Gl. 8. Lucia Stream 9. Terrace Point 10. Strawberry Island 11. Atrevida Gl. 12. Ampitheater Knob 13. Esker Stream 14. Galianao Gl. 15. Black Gl. 16. Turner Gl. 17. Haenke Gl. 18. Hubbard Gl. 19. Osier Island 20. Gilbert Point 21. Haenke Island 22. Marble Point 23. Mt. Alexander 24. Alexander Gl. 25. Indian Camp 26. Logan Beach 27. Knight Island 28. Otmeloi Island 29. Khantaak Island 30. Yakutat 31. Cape Stoss 32. Cape Enchantment (b) Sites visited in Glacier Bay. Numbered locations are: 1. Rendu Gl. 2. Rendu Inlet 3. Reid Gl. 4. Hugh Miller Gl. 5. Charpentier

Gl. 6. Geike Gl. 7. Wood Gl. 8. Carroll Gl. 9. Morse Gl. 10. Muir Gl. 11. McBride Gl. 12. Casement Gl. 13. Tidal Inlet 14. Muir Inlet 15. Herbert Gl. 16. Mendenhall Gl. 17. Norris Gl. 18. Taku Gl. 19. Russell Island 20. Triangle Island 21. N. Marble Island

Figure 3. Sites visited in Greenland. (a) Site locations in the Qaanaaq region. (b) Site locations in the Upernavik

Archipelago region. (c) Site locations in the Disko Bay region.

Figure 4. Examples of photographs of glacial features. (a) Push moraine at front of Columbia Glacier (note person for scale) (ID: tve_lanternslide_0004). (b) Kettles in outwash plain of Hidden Glacier (ID: tve_exp1905_162).

Figure 5. Photographs showing effects of 1899 earthquakes. (a) Wave cut bench and sea cliff on east shore of Haenke Island

uplifted during earthquakes. (ID: tve_exp1906_183). (b) Trees on Khantaak Island killed by submergence in salt water due to earthquakes (ID: tve_exp1909_325).

Figure 6. Equipment and field conditions. (a) Camera used in the Alaska expeditions. (b) Close up of camera front. (c) Expedition party traversing glacier in Alaska. (d) Expedition member with photography gear traversing cliff. (e) Scientific party and camera set up, Russell Fjord, Alaska. (ID: tve_exp1909_002a). (f) Example of panoramic image, Columbia

Glacier, Alaska (ID: tve_exp1909_035).

Figure 7. Digitized images from Greenland. (a) Icebergs in harbor of Umanak (Uummannaq) (ID: tve_lanternslide_0173). (b) Terminus of Nugsuak Glacier. (ID: tve_exp1896_147). (c) Icebergs in Waigat Strait. (ID: tve_exp1986_186). (d) North Cornell Glacier. (ID: tve_lanternslide_0090).



Figure 8. Digatized images from the 1905 Alaska expedition. (a) Hubbard Glacier from Osier Island (ID: tve_exp1905_100).

(b) Terminus of Nunatak Glaicer (ID: tve_lanternslide_0264).

Figure 9. Digitized images from the 1906 Alaska expedition. (a) Variegated Glacier from Gilbert Point (ID: tve_exp1906_219). (b) Turner and Haenke Glaciers from Haenke Island (ID: tve_exp1906_205_02. (c) Turner Glacier from Gilbert Point (ID: tve_exp1906_213). (d) Hidden Glacier and outwash plain (ID: tve_lanternslide_0267).

Figure 10. Digitized images from the 1909 Alaska Expedition. (a) Shoup Glacier terminus. (ID: tve_exp1909_032). (b)

Miles Glacier and Copper River Railroad. (ID: tve_exp1909_041).

Figure 11. Digitized images from the 1911 Alaska expedition. (a) Muir Glacier. (ID: tve_lanternslide_0025). (b) Street in Dawson, Yukon. (ID: tve_exp1911_042). (c) Effort to build diversion dam for Spencer Glacier Stream. (ID: tve_exp1911_030). (d) Tracks displaced by Spencer Glacier outlet stream. (ID: tve_exp1911_020).

Figure 12. Digitized images from the Ithaca area. (a) Ridge of an esker, McLean, NY (ID: tve_ithaca_07). (b). Taughannock

Falls in 1888. (ID: tve_ithaca_02).






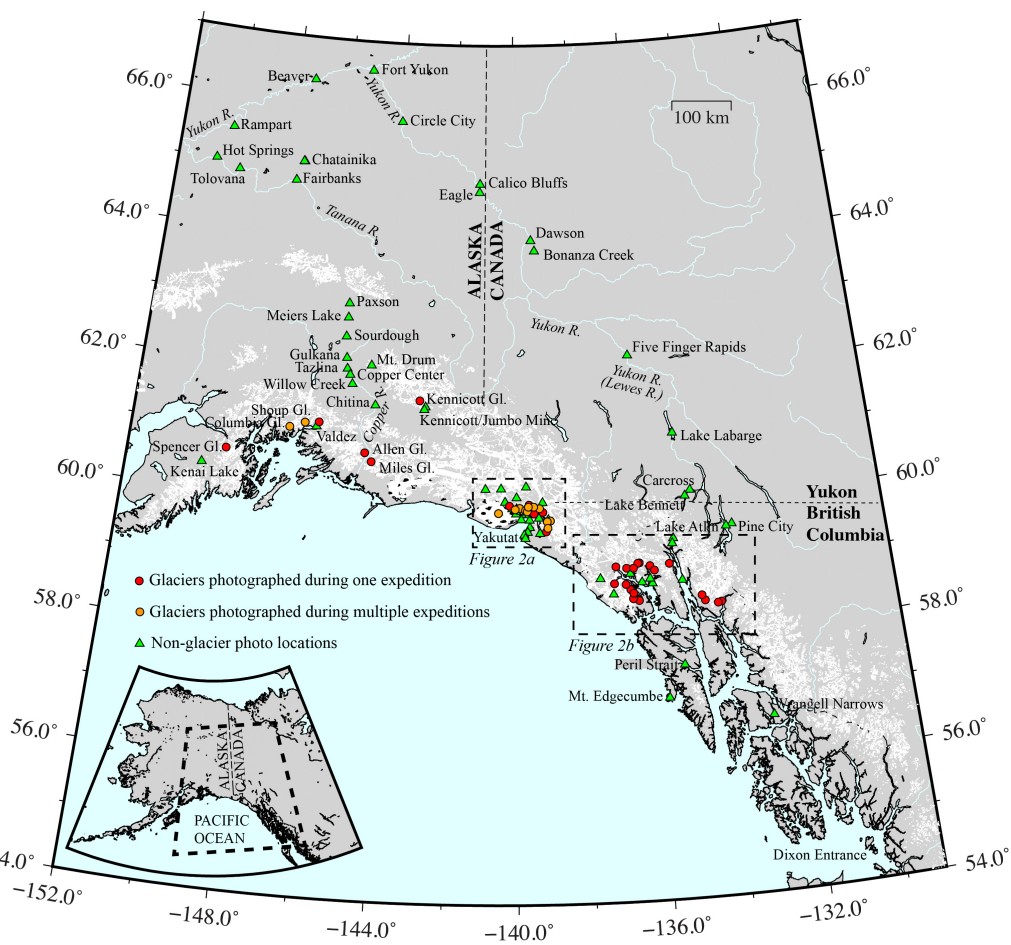

Figure 1.






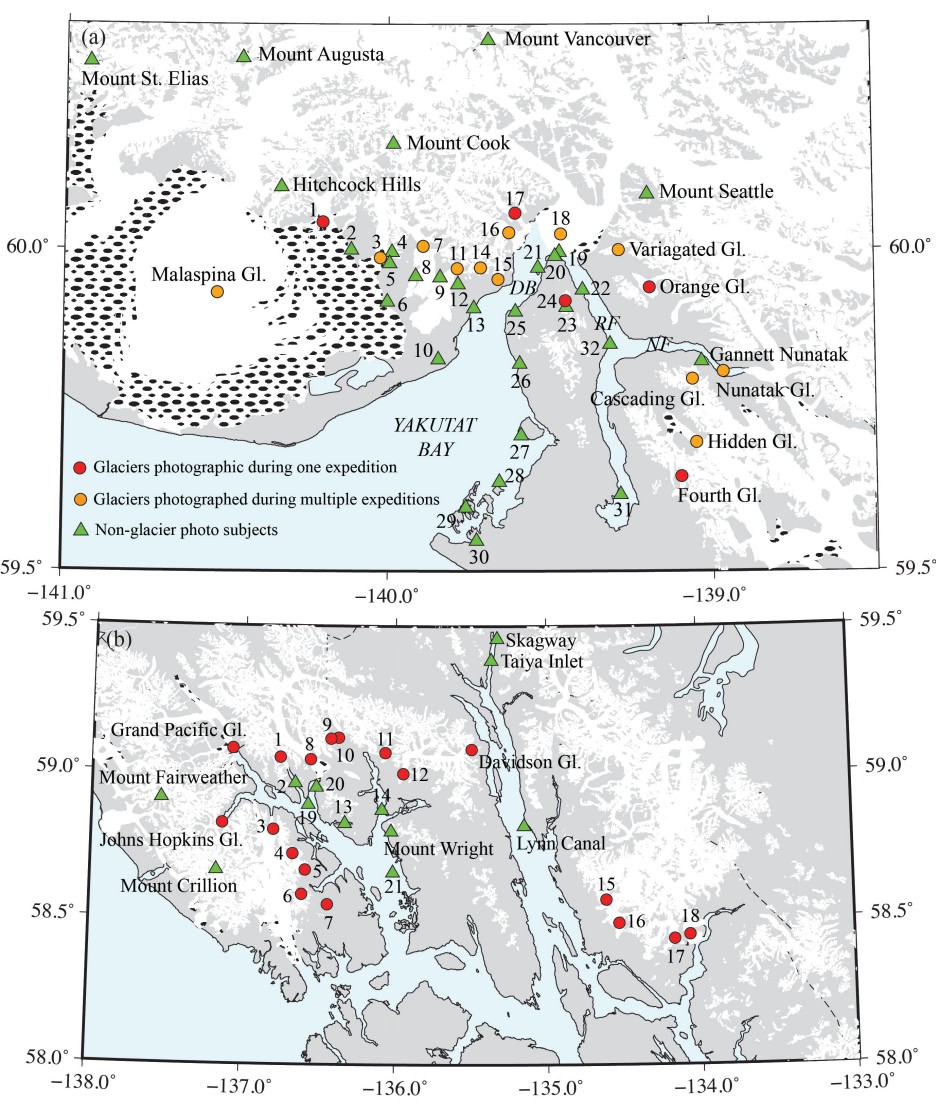

Figure 2.





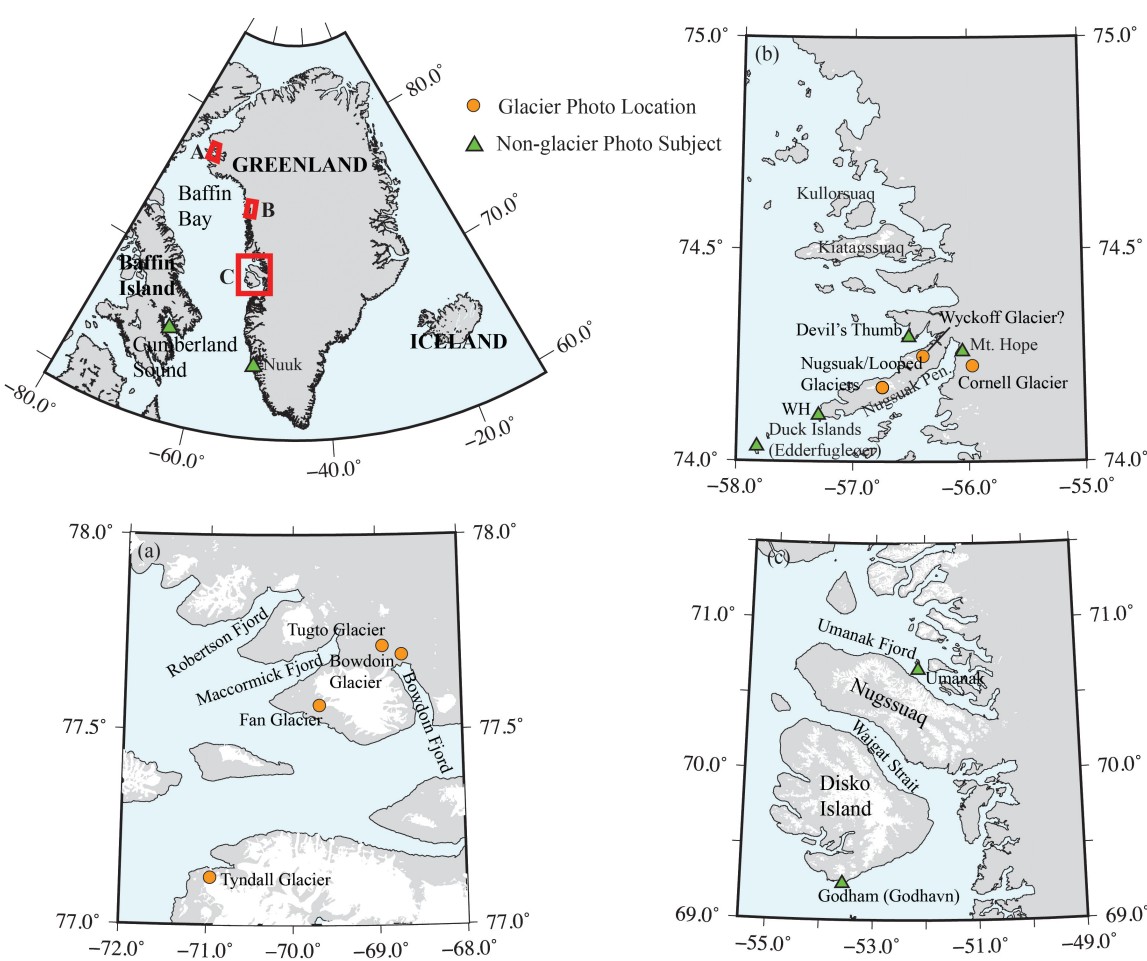

Figure 3.






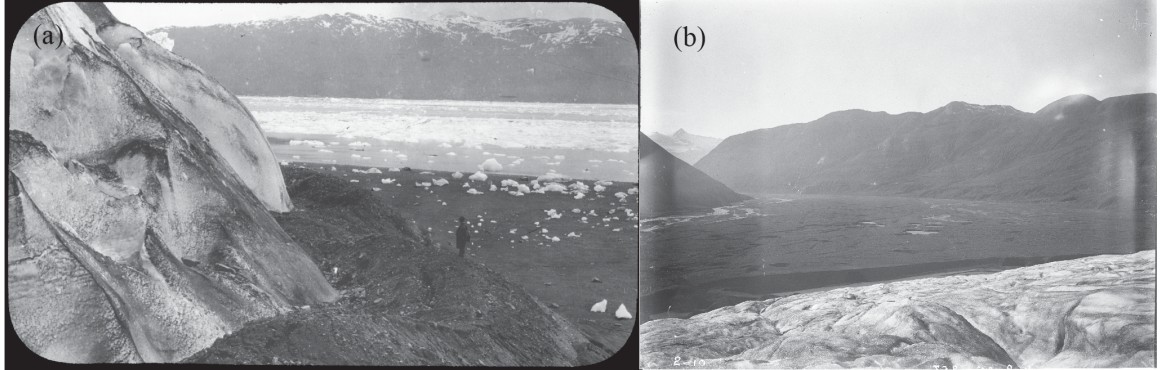

Figure 4.


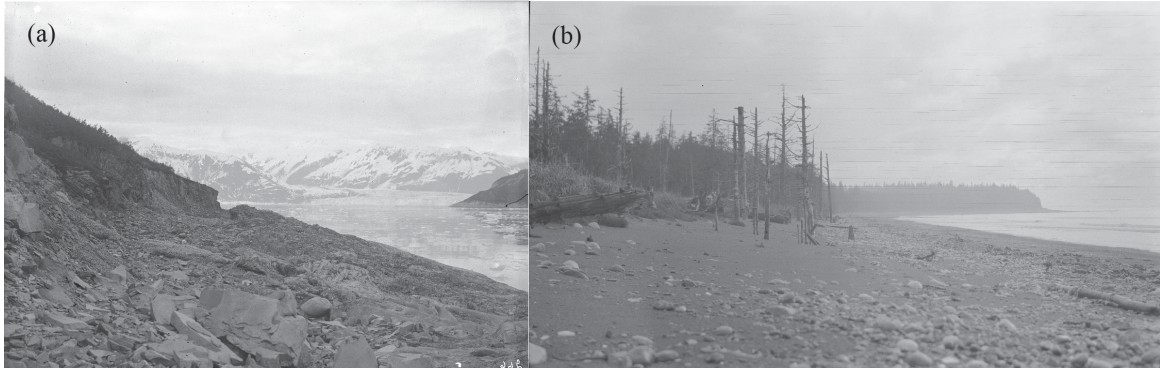

Figure 5.




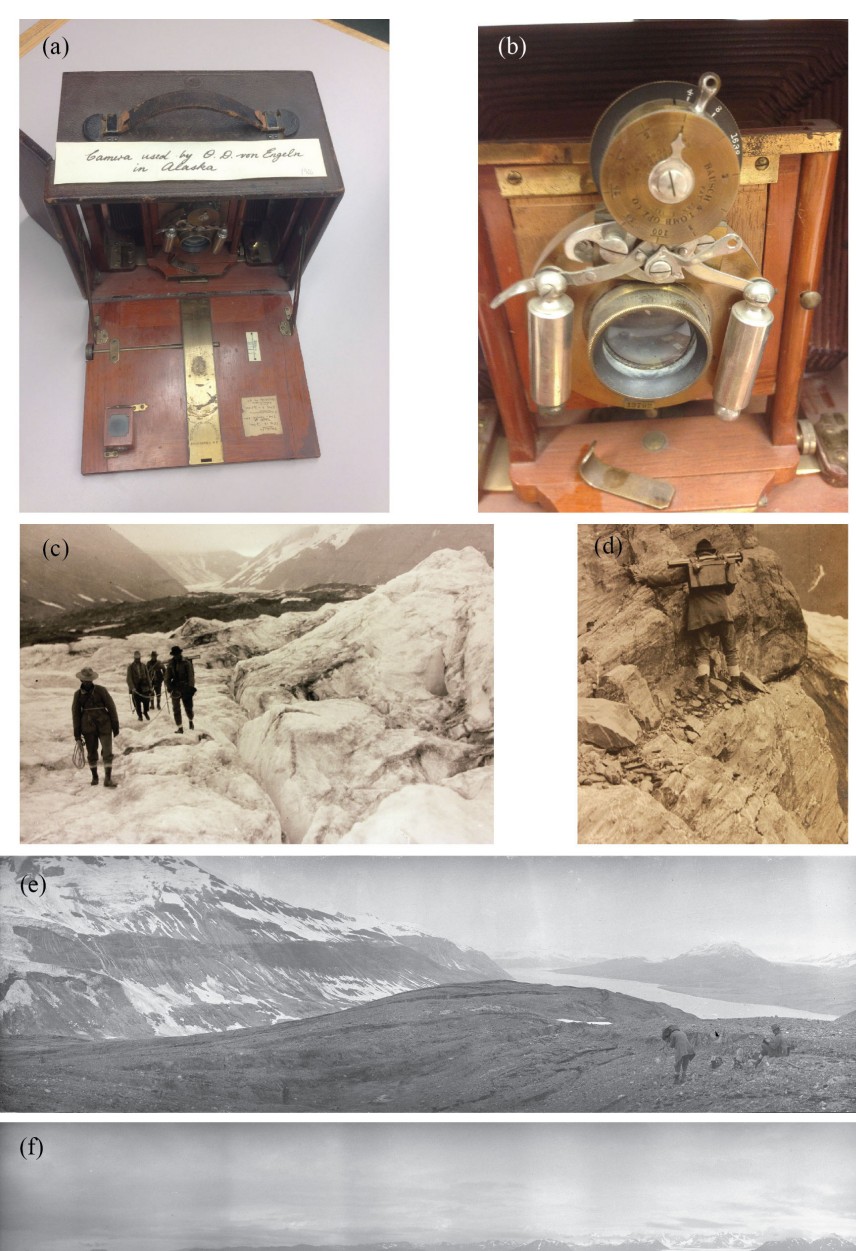

Figure 6.





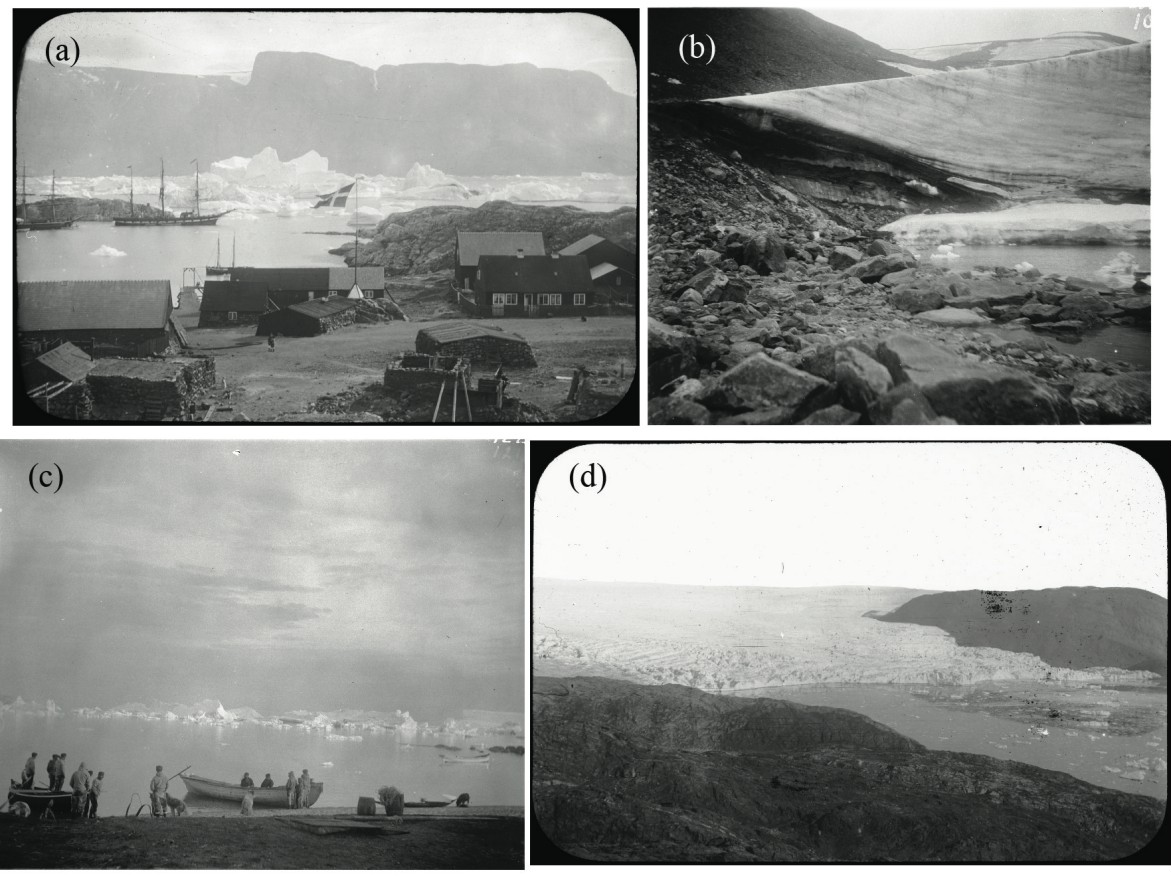

Figure 7.




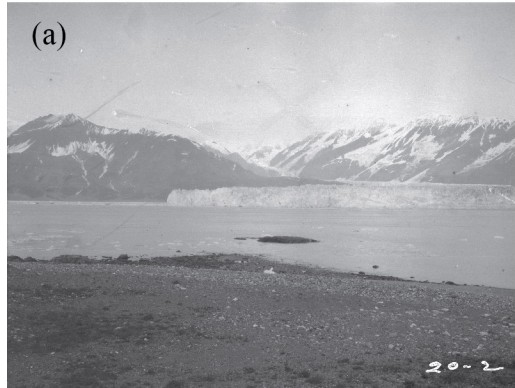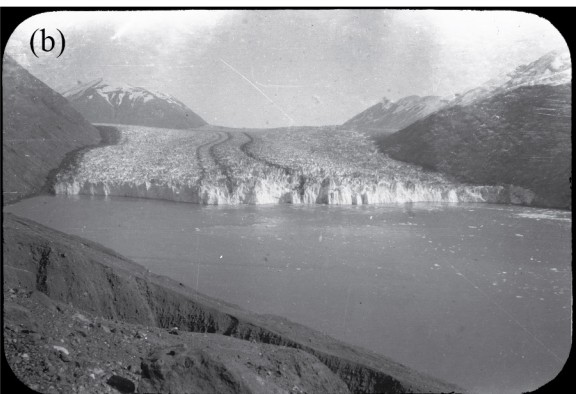

Figure 8.







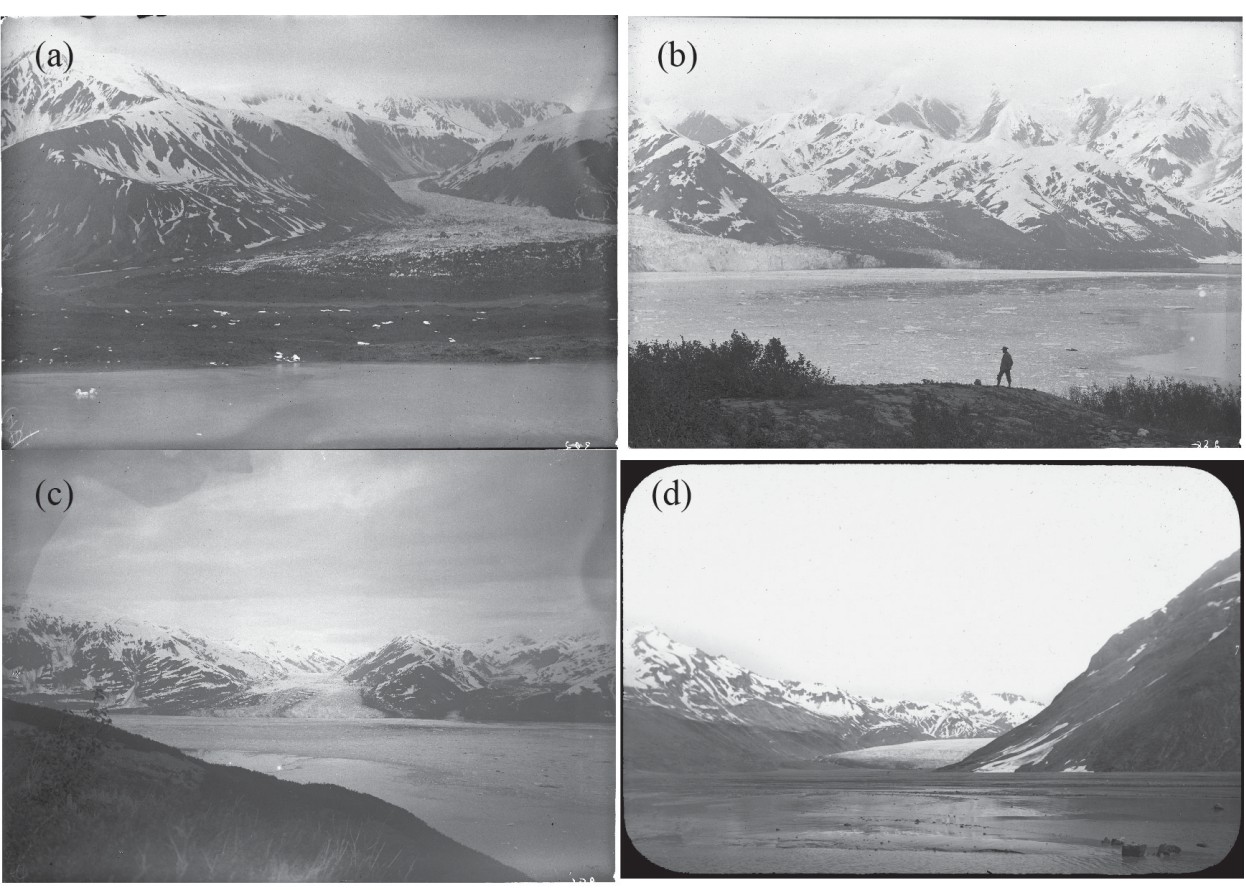

Figure 9.





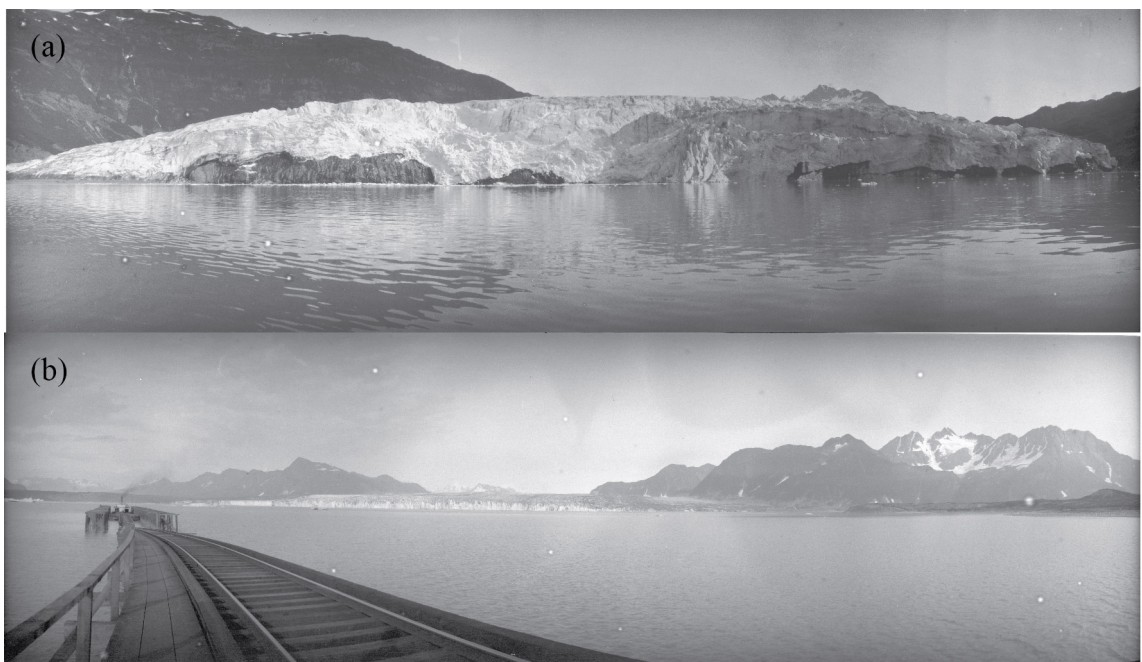

Figure 10.






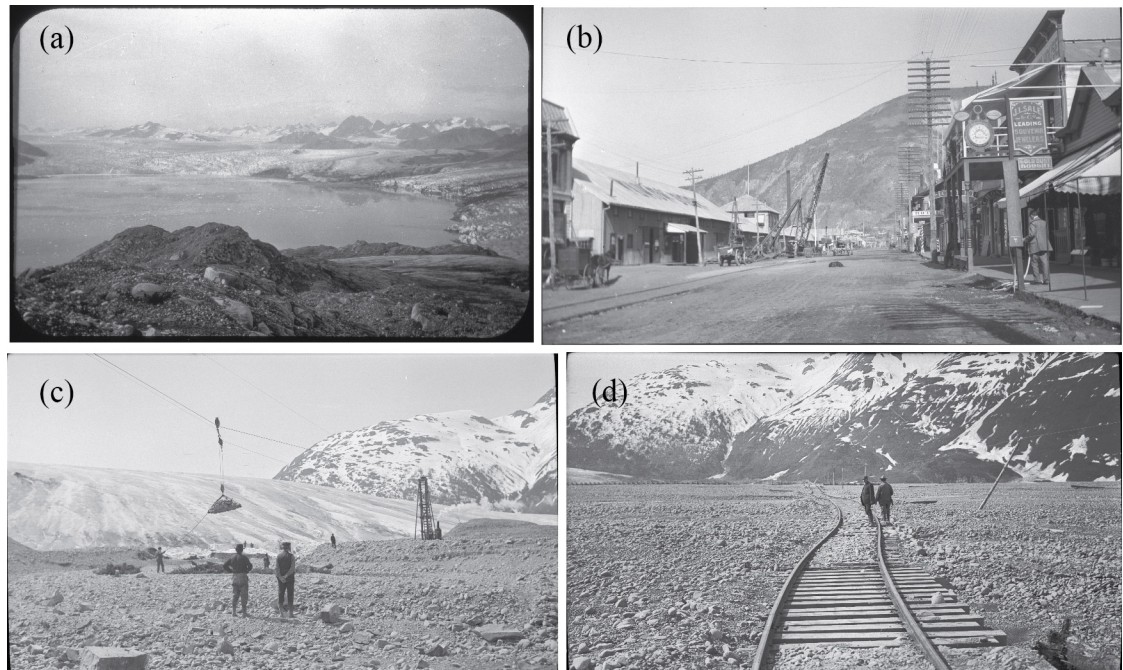

Figure 11.



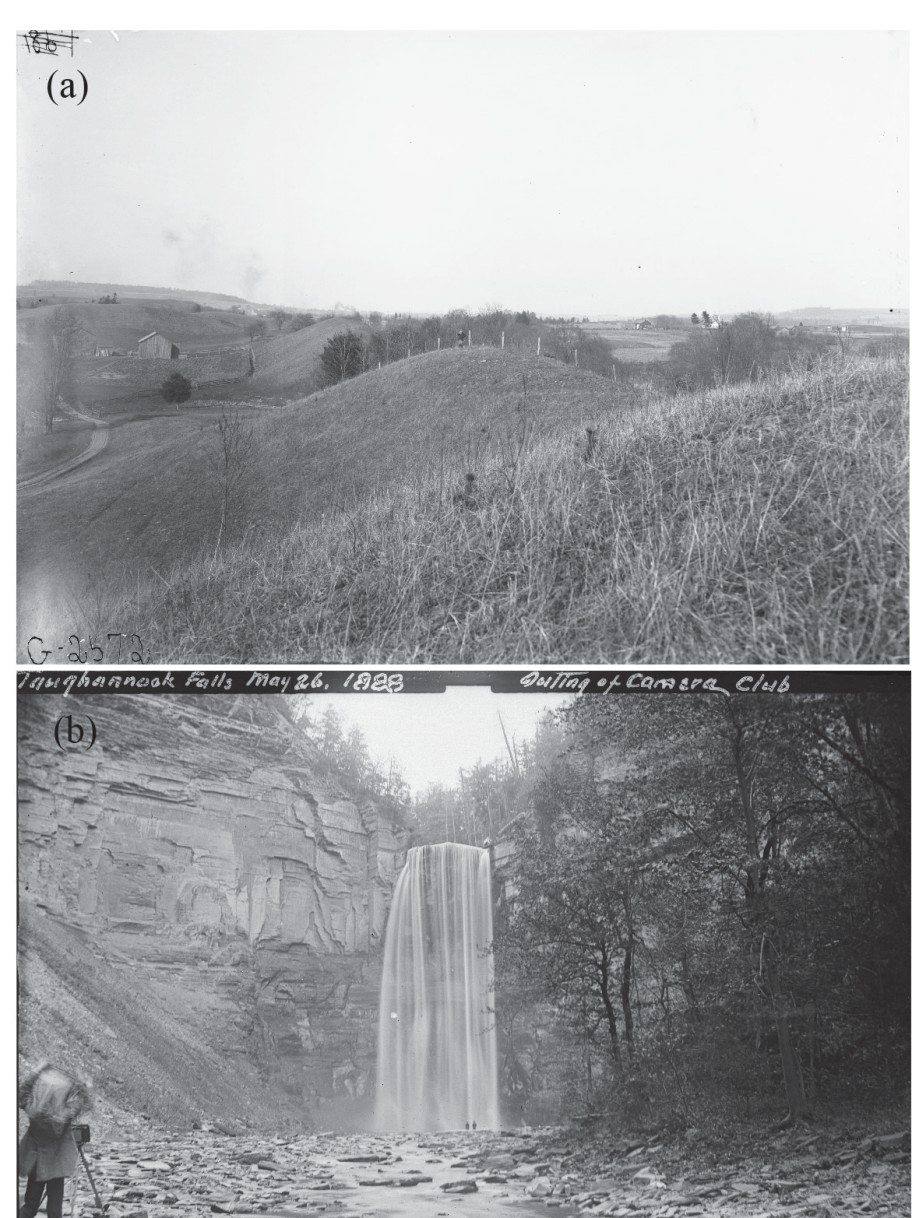


Figure 12.



| Expedition/Year | Approximate number of photos | Examples of key geographic locations |
|---|---|---|
| 1896 | 374 | Labrador and Baffin Land, Canada; Nugsuak Peninsula, Waigat Strait, Disko Island, and Umenak Fjord, Greenland |
| 1905 | 198 | Yakutat Bay and steamer ride from Seattle |
| 1906 | 264 | Yakutat Bay and steamer ride from Seattle |
| 1909 | 582 | Yakutat Bay, eastern Prince William Sound, Kenai Peninsula, lower Copper River, coastal southeast Alaska , Inside Passage to Seattle |
| 1911 | 356 | Prince William Sound, Kenai Peninsula, Wrangell Mountains, Interior Alaska (including Fairbanks and several mining towns), Yukon Territory, northern British Columbia, Glacier Bay, Inside Passage to Seattle |
| Lantern slides/ various years | 308 | Alaska, Canada, Greenland, Antarctica. |
| Ithaca/ 1888-1928 | 54 | Waterfalls and glacial landforms around Ithaca, New York |

Table 1. Summary of digitized photographs.



| Glacier | Region | Years of Photos in Collection | # of Photos in Collection | # of Photos of Terminus[**] | Multiple Viewpoints[^^] | Large Area[##] |
|---|---|---|---|---|---|---|
| Columbia | Prince William Sound | 1909, 1911 | 79 | 12 | Yes | Yes |
| Spencer | Prince William Sound | 1911 | 40 | 5 | No | No |
| Shoup | Prince William Sound | 1909, 1911 | 33 | 6 | Yes | Yes |
| Valdez | Prince William Sound | 1909 | 30 | 5 | No | No |
| Catspaw* | Prince William Sound | 1909 | 1 | - | - | - |
| Miles | Chugach Mts. | 1909 | 17 | 4 | No | No |
| Allen (Baird)^ | Chugach Mts. | 1909 | 13 | - | - | - |
| Kennicott | Wrangell Mts. | 1911 | 15 | - | - | Yes |
| Hubbard | Yakutat | 1905, 1906, 1909 | 82 | 28 | Yes | Yes |
| Turner | Yakutat | 1905, 1906, 1909 | 82 | 16 | Yes | No |
| Galiano | Yakutat | 1905, 1906, 1909 | 16 | - | - | - |
| Atrevida | Yakutat | 1905, 1906, 1909 | 51 | 2 | No | Yes |
| Variegated | Yakutat | 1905, 1906, 1909 | 44 | 4 | No | No |
| Cascading | Yakutat | 1905, 1906, 1909 | 6 | - | - | - |
| Nunatak | Yakutat | 1905, 1906, 1909 | 82 | 28 | Yes | Yes |
| Hayden | Yakutat | 1905, 1906, 1909 | 27 | - | - | - |
| Malaspina | Yakutat | 1905, 1906, 1909 | 24 | 2 | No | Yes |
| Lucia | Yakutat | 1905, 1906, 1909 | 30 | 5 | No | Yes |
| Hidden | Yakutat | 1905, 1906, 1909 | 79 | 26 | Yes | Yes |
| Marvine | Yakutat | 1906 | 31 | - | - | - |
| Alexander | Yakutat | 1906 | 1 | - | - | - |
| Orange | Yakutat | 1906 | 7 | 1 | No | No |
| Fourth (Beasley)# | Yakutat | 1909 | 18 | 5 | Yes | No |
| Black | Yakutat | 1905, 1909 | 2 | - | - | - |
| Haenke | Yakutat | 1909 | 4 | 2 | No | No |
| Flat* | Yakutat | 1905 | 4 | - | - | - |
| Serpentine* | Yakutat | 1905 | 3 | - | - | - |
| Fallen‡ | Yakutat | 1905 | 1 | - | - | - |
| Muir | Glacier Bay | 1911 | 28 | 6 | Yes | Yes |
| McBride | Glacier Bay | 1911 | 4 | 1 | No | No |
| Casement | Glacier Bay | 1911 | 1 | - | - | - |
| Morse | Glacier Bay | 1911 | 1 | 1 | No | No |
| Rendu | Glacier Bay | 1911 | 11 | 4 | No | No |
| Wood | Glacier Bay | 1911 | 2 | 2 | No | No |
| Davidson | Glacier Bay | 1911 | 4 | 2 | No | No |
| Hugh Miller | Glacier Bay | 1911 | 8 | 3 | No | No |
| Carroll | Glacier Bay | 1911 | 3 | 2 | No | No |
| Grand Pacific | Glacier Bay | 1911 | 2 | 2 | No | No |
| Johns Hopkins | Glacier Bay | 1911 | 3 | - | - | - |
| Reid | Glacier Bay | 1911 | 2 | 2 | No | No |
| Charpentier | Glacier Bay | 1911 | 6 | 3 | No | No |
| Geikie | Glacier Bay | 1911 | 2 | 1 | No | No |
| White§ | Glacier Bay | Not listed | 1 | 1 | No | No |
| Favorite | Glacier Bay | 1911 | 3 | - | - | - |
| Girdled§ | Glacier Bay | Not listed | 1 | | | |
| Taku | Juneau | 1911 | 7 | 4 | No | No |
| Norris | Juneau | 1911 | 3 | - | - | - |
| Herbert | Juneau | 1911 | 1 | - | - | - |
| Mendenhall | Juneau | 1911 | 1 | - | - | - |

* Location and identification uncertain

^ Tarr referred to this glacier as Baird; the later accepted name is Allen

# Tarr used both names for this glacier in the photos; the accepted name is Fourth

‡ Tarr and Martin (1914) reported that this glacier slid into the Disenchantment Bay on July 4, 1905, causing a local tsunami

§ Glacier only appears in a photo by H. Reid on a lantern slide

** To be counted in this category, the photo had to show the majority of the terminus region

^^ At least three viewpoints of terminus region (e.g. front view and angled from each side)

## Photos (either as a single photo or collectively) show at least 5 km of the glacier from the terminus region

Table 2. Alaska Glacier Photographs in Collection





| Glacier* | Region | Year of Photos | # of Photos | # of Photos of Terminus[**] | Multiple Viewpoints[^^] | Large Area[##] |
|---|---|---|---|---|---|---|
| Cornell *(Ikissuup Sermersua)* | Nuussuaq Peninsula - Upernavik Archipelago | 1896 | 88 | 11 | No | No |
| Wyckoff | Nuusssuaq Peninsula - Upernavik Archipelago | 1896 | 3 | - | - | - |
| Nugsuak[#] *(Sermikavsak?)* | Nuussuaq Peninsula - Upernavik Archipelago | 1896 | 12 | - | - | - |
| Bowdoin^ (*Kangerluarsuup Sermia*) | Qaanaaq | 1894 or 1899 | 10 | 1 | No | Yes |
| Tuktoo (Tugto)^ (*Tuttu*) | Qaanaaq | 1894 or 1899 | 1 | - | - | - |
| Fan^ *(Fan)* | Qaanaaq | 1894 or 1899 | 1 | - | - | - |
| Robertson^ | Qaanaaq | 1894 or 1899 | 1 | 1 | No | No |
| Tyndall^ *(Tyndall)* | Qaanaaq | 1894 or 1899 | 1 | 1 | No | No |

* Regular type gives name of glacier as known to the photographer at the time the image was acquired. Italic type gives Greenlandic name or name it is commonly known as today.
[#]Variably spelled as Nugsuak or Nugssuak in collection materials
^ Only appears as photos by Libbey on lantern slides
** To be counted in this category, the photo had to show the majority of the terminus region
^^ At least three viewpoints of terminus region (e.g. front view and angled from each side)
[##] Photos (either as a single photo or collectively) show at least 5 km of the glacier from the terminus region

Table 3. Greenland Glacier Photographs in Collection