# Peer review of "Historic photographs of glaciers and glacial landforms from the R. S. Tarr collection at Cornell University"

_Earth System Science Data, 2019_

## Referee Comment (RC1) · Anders Anker Bjørk (Referee) · 25 Jun 2019

Elliott and Pritchard have brought forward valuable knowledge on a unique archive of historical glacier images. There is no doubt that this paper is very useful and will raise the awareness and use of the images. In terms of scientific data, the images are raw and will need to be further processed to be useful in scientific studies. It is great that the authors have put this manuscript together, and their work serves as an example that I hope many more archives will follow.

I have a few comments to the manuscript:

[Figure]

1) It is not clear to me why the images need to be in two different online archives – each with different level of metadata and (apparently also) different file sizes. I will recommend that you only link to the archive with the most metadata and the largest file sizes/original scans.

2) I was not able to find the original scan sizes in ARTSTOR, here the images were also c. 1.5 mb as in the Cornell Library. For a 4x5 inch glass plate scanned at high resolution this is very downsized. It would be great with a link or DOI for the high resolution images.

3) From you figures it shows that many of your images have already been geolocated – this information is not transferred to the metadata. I recommend that this location data is added to the metadata, or alternatively a supplementary file with image coordinates is attached to the manuscript.

4) In your manuscript you call it the R.S. Tarr Collection – in the online archives it is called "Historic Glacial Images of Alaska and Greenland". I find this title misleading as there are images from other locations and other subjects than glaciers too.

5) The historical images in your figures do not appear to do the material justice. I would like to see a better processing of the photos with more contrast, also the cropping and set-up of the photos can be improved.

6) Lastly, I am looking more for information on the completeness of the Tarr collection. Are there other images/material from the expeditions in other archives?

With all this said, I am very happy with the manuscript, and hope to see it published and the images used for science again!
* * *

---

## Referee Comment (RC2) · Florence Fetterer (Referee) · 31 Aug 2019

General comments

This is a well-presented description of a valuable collection of newly digitized photographs. The data set of digital images is accessible and reasonably well described with searchable metadata. With publication of this paper, the number of users of the photograph collection will increase, and increasing use is fundamental to preserving historical collections.

Specific comments

[Figure]

This was a satisfying paper to read for several reasons. It is well written in plain language, it documents progress in preserving and promoting the use of important historical data, and it points to how archivists and geophysicists can collaborate. The paper lacks some detail on how metadata fields are chosen and how metadata is stored that would be helpful for those contemplating similar work.

20-24: Various interest groups will be able to use the images for purposes beyond geophysical research. It's good to see that fact laid out clearly and prominently in these lines. The authors may want to reference Maness et al., 2017, doi:10.1016/j.grj.2017.10.002 because it stresses the importance of this kind of work through several examples, of which the NSIDC Glacier Photograph Collection is the most prominent.

45 . . .A summary of the full collection is shown in Table 1. . . .

It's not clear what the full collection referred to here is. It is the digitized collection of photographs, but those are part of a larger collection, and later it says

200 . . .Due to budget constraints, we could not digitize all of the images, but focused on images of glaciers and glaciated landscapes from the Alaska and Greenland expeditions. There are still hundreds of other photographs in the RMC that were not digitized . . .

Were all of the Alaska and Greenland expeditions photos digitized? Please clarify earlier in the manuscript (earlier than line 200) if this is so. "Collection" has a particular meaning in library archives, so referring to "full collection" can be confusing without some additional descriptive words.

48-58: Tables 2 and 3, with basic information on the photos in the digital collection, are really helpful for giving the reader a quick way to assess how useful the collection may be for their work. It is a good example for other collections to follow. NSIDCs Glacier Photograph Collection is too big to tabulate in exactly the same way, but we aspire to
having searchable metadata that would allow such a table to be easily built.

186-187: . . .Over a forty year period (including 20 years at Cornell), Tarr accumulated images of glacial landforms ..

And he died at 48? This implies that he had been collecting photographs since he was 8, which is certainly possible, but if so, it is so remarkable that it would be good to add some comment about it!

Section 3.1 on Original Material, lines 195-203: At NSIDC we have a similar situation. As of August 2018, the analog glacier photographs reside not at NSIDC, but in the University of Colorado at Boulder (CU) Library Archive. The Digital Library at CU and NSIDC collaborated on improving the digital portion of the collection, and both NSIDC and CU Library maintain interfaces to and identical copies of the digital portion of this large analog collection. These interfaces serve different user communities. The Background section of the User Guide, under https://nsidc.org/data/g00472, describes some of this and links to the CU Digital Library collection interface at https://cudl.colorado.edu/luna/servlet/CUB~12~12). At some point in the manuscript, it may be useful to link to the CU Library interface as well as to the NSIDC interface to the digital collection.

212-213: . . .. Master images were saved as 16B layered tiffs with curve adjustment layers. Uploaded access files are flattened, 8B, high quality compression, original sized jpegs. . . .

Do the above lines mean that 16 bit images were digitally archived, but only 8 bit images are accessible/distributed? If so, perhaps some words on why 8 bit serve as well as 16 bit would be helpful.

221: Please list the range of file sizes, in MB, for the images, along with typical size.

224: Did you consider giving not only the entire collection but each individual photograph image a DOI?

Section 3.4 Metadata and Filenames

I found this section somewhat hard to follow. Adding some example filenames (e.g. "tve_lanternslide_0007.jpg" would help. Adding a listing of all the metadata fields that were used, along with the range of possible values for each field, would be good. It would also be good to make the metadata for the collection available as a downloadable spreadsheet. Was this considered? As it is, the only way I see of using the metadata is to make selections from the user interface. That's adequate for most users, but having the ability to download all metadata would be a great way to get a complete and detailed picture of the collection for those who want to go further.

Consider including a figure that is a screenshot of a page like this, https://digital.library.cornell.edu/catalog/ss:9417838 , so that the discussed metadata fields are shown with a photograph, as it would appear to a data collection user.

259-265: . . .In some cases, the envelope includes some "additional notes" that have not been included in the digital published metadata through the Cornell Library, but that are included in Table S1. For example, some images included a letter grade for the quality of the image (A+ being the highest and D being the lowest), presumably assessed by Tarr or von Engeln. In some other images, these notes include the name of the photographer, or if the photograph is duplicated as a lantern slide set, the number of the lantern slide is given (see above). Table S1 also includes the appropriate USGS Quadrangle topographic map or Natural Resources Canada topographic map for the images when available. . ..

This (above) is the first mention of Table S1. Please define it and put it in context. I am guessing it is a table of supplemental metadata. Is it accessible?

Lines 267 – 285: The detail is great, and it helps readers understand how complicated it can be to assign metadata consistently. But again, I had trouble following this section, and think a listing of all metadata fields, in a table with example rows, would help.

283: "Twenty-four photographs from Martin's trips...". Searching now, I find 35 attributed to L. Martin, and 15 to an E.R. Martin, all from Alaska in the early 1900s.

Would you please cite the NSIDC collection somewhere in this section as "National Snow and Ice Data Center (2015)" and then have this, below, in the reference section?

National Snow and Ice Data Center (comp.). 2002, updated 2015. Glacier Photograph Collection, Boulder, Colorado USA. NSIDC: National Snow and Ice Data Center. doi: https://doi.org/10.7265/N5/NSIDC-GPC-2009-12. [Date Accessed].

294: I am happy to report that the NSIDC glacier photo collection now has more than 24,000 glaciers on line. More were digitized and described thanks to a grant to the CU Library (see Maness et al. 2017)

295 ...The NSIDC also holds and has scanned Martin's field notebooks from the 1904, 1905, and 1906 Alaska expeditions.

Are you able to give me a source or a link for the scanned Martin notebooks? NSIDC has not had an archivist for more than five years now, and I am not able to find the scans you reference. I believe the physical notebooks are safe with CU Library.

Finally, I agree with RC1, who writes " From you figures it shows that many of your images have already been geolocated – this information is not transferred to the metadata. I recommend that this location data is added to the metadata, or alternatively a supplementary file with image coordinates is attached to the manuscript. "

Technical corrections

28: correct typo and formatting in Hill et al.

102: think it needs a paragraph break after this line.

108: missing a close paren.

221: typo "originally size"

Thank you for the opportunity to review this manuscript. It has been a pleasure.

---

## Author Comment (AC1) · 4 Nov 2019

We would like to thank Florence Fetterer for her time and detailed review of the manuscript. In our response, the reviewer comments are shown in black while our answers are shown in blue. We also reference the relevant lines in the revised manuscript, in which changes are highlighted.

20-24: Various interest groups will be able to use the images for purposes beyond geophysical research. It's good to see that fact laid out clearly and prominently in these lines. The authors may want to reference Maness et al., 2017, doi:10.1016/j.grj.2017.10.002 because it stresses the importance of this kind of work through several examples, of which the NSIDC Glacier Photograph Collection is the most prominent.

This reference has been added – thank you for the suggestion.

A summary of the full collection is shown in Table 1.

It's not clear what the full collection referred to here is. It is the digitized collection of photographs, but those are part of a larger collection, and later it says

Due to budget constraints, we could not digitize all of the images, but focused on images of glaciers and glaciated landscapes from the Alaska and Greenland expeditions. There are still hundreds of other photographs in the RMC that were not digitized

Were all of the Alaska and Greenland expeditions photos digitized? Please clarify earlier in the manuscript (earlier than line 200) if this is so. "Collection" has a particular meaning in library archives, so referring to "full collection" can be confusing without some additional descriptive words.

We have clarified the wording by changing "full collection" to "digitized collection" (line 47 in the revised manuscript). The priority for digitization was Alaska and Greenland photos, then Ithaca region images. To the best of our knowledge, close to all of the Alaska and Greenland images with sufficient original material quality were digitized. We cannot guarantee that some images were not overlooked (some materials were not well organized). Some images were not digitized due to the original material being in poor shape, some others that were duplicates of already scanned images were also skipped over. We are working with Cornell library staff to obtain firm numbers of remaining, unscanned images. We have added more discussed of the remaining images (lines 208-213 in revised manuscript).

Over a forty year period (including 20 years at Cornell), Tarr accumulated images of glacial landforms ..
And he died at 48? This implies that he had been collecting photographs since he was 8, which is certainly possible, but if so, it is so remarkable that it would be good to add some comment about it!

This was a typo and has been fixed (line 190 in revised manuscript).

Section 3.1 on Original Material, lines 195-203: At NSIDC we have a similar situation. As of August 2018, the analog glacier photographs reside not at NSIDC, but in the University of Colorado at Boulder (CU) Library Archive. The Digital Library at CU and NSIDC collaborated on improving the digital portion of the collection, and both NSIDC and CU Library maintain interfaces to and identical copies of the digital portion of this large analog collection. These interfaces serve different user communities. The Background section of the User Guide, under https://nsidc.org/data/g00472, describes some of this and links to the CU Digital Library collection interface at https://cudl.colorado.edu/luna/servlet/CUB. At some point in the manuscript, it may be useful to link to the CU Library interface as well as to the NSIDC interface to the digital collection.

A link to the CU libraries has been added (line 359 in revised manuscript).

Master images were saved as 16B layered tiffs with curve adjustment layers. Uploaded access files are flattened, 8B, high quality compression, original sized jpegs.

Do the above lines mean that 16 bit images were digitally archived, but only 8 bit images are accessible/distributed? If so, perhaps some words on why 8 bit serve as well as 16 bit would be helpful.

The full resolution, 16 bit TIFF files are now available to download through the Cornell library interface.  The standard policy for the Cornell libraries is to archive the full resolution images and make smaller files available for download, which is why only the smaller files were available initially.

221: Please list the range of file sizes, in MB, for the images, along with typical size.

There is not a simple way for library staff to assess the range of the file sizes, but the average size is 135 MB for the full resolution TIFFs.

224: Did you consider giving not only the entire collection but each individual photograph image a DOI?

We did not initially consider assigning individual DOIs.  We are now exploring practicalities and options with library staff as adding ~2000 DOIs would be a non-trivial task.  There are pros and cons to individual DOIs.  If someone used a couple of the images, individual DOIs would be a benefit but if someone was using hundreds of the images it could become cumbersome.

Section 3.4 Metadata and Filenames
I found this section somewhat hard to follow. Adding some example filenames (e.g. "tve_lanternslide_0007.jpg" would help. Adding a listing of all the metadata fields that

were used, along with the range of possible values for each field, would be good. It would also be good to make the metadata for the collection available as a downloadable spreadsheet. Was this considered? As it is, the only way I see of using the metadata is to make selections from the user interface. That's adequate for most users, but having the ability to download all metadata would be a great way to get a complete and detailed picture of the collection for those who want to go further.
Consider including a figure that is a screenshot of a page like this, https://digital.library.cornell.edu/catalog/ss:9417838 , so that the discussed metadata fields are shown with a photograph, as it would appear to a data collection user.

A spreadsheet with all the metadata fields was available as part of the supplemental information (which was in a separate file from the manuscript). An updated version of this spreadsheet (with additional metadata fields, including approximate coordinates) is included in the supplemental information to this revised submission. The spreadsheet (Table S1) includes a listing of all the metadata fields as well as the full range of values for each field. We have moved the first mention of S1 to the first sentence of the section to make the table more prominent to the reader.

An example filename is given (line 258 in revised manuscript).

We have also reorganized parts of the discussion of the metadata and filename section to make it clearer.

In some cases, the envelope includes some "additional notes" that have not been included in the digital published metadata through the Cornell Library, but that are included in Table S1. For example, some images included a letter grade for the quality of the image (A+ being the highest and D being the lowest), presumably assessed by Tarr or von Engeln. In some other images, these notes include the name of the photographer, or if the photograph is duplicated as a lantern slide set, the number of the lantern slide is given (see above). Table S1 also includes the appropriate USGS Quadrangle topographic map or Natural Resources Canada topographic map for the images when available

This (above) is the first mention of Table S1. Please define it and put it in context. I am guessing it is a table of supplemental metadata. Is it accessible?
Lines 267 – 285: The detail is great, and it helps readers understand how complicated it can be to assign metadata consistently. But again, I had trouble following this section, and think a listing of all metadata fields, in a table with example rows, would help.

See above.

283: "Twenty-four photographs from Martin's trips...". Searching now, I find 35 attributed to L. Martin, and 15 to an E.R. Martin, all from Alaska in the early 1900s.

We have corrected our numbers (line 354 in revised manuscript).

Would you please cite the NSIDC collection somewhere in this section as "National Snow and Ice Data Center (2015)" and then have this, below, in the reference section? National Snow and Ice Data Center (comp.). 2002, updated 2015. Glacier Photograph Collection, Boulder, Colorado USA. NSIDC: National Snow and Ice Data Center. doi: https://doi.org/10.7265/N5/NSIDC-GPC-2009-12. [Date Accessed].

Done.

294: I am happy to report that the NSIDC glacier photo collection now has more than 24,000 glaciers on line. More were digitized and described thanks to a grant to the CU Library (see Maness et al. 2017)

We have changed our mention of the NSIDC holdings to reflect the updated number (line 355 in revised manuscript).

295 ...The NSIDC also holds and has scanned Martin's field notebooks from the 1904, 1905, and 1906 Alaska expeditions.

Are you able to give me a source or a link for the scanned Martin notebooks? NSIDC has not had an archivist for more than five years now, and I am not able to find the scans you reference. I believe the physical notebooks are safe with CU Library.

This is correct – the notebooks are in the CU Library. We have changed the mention in Section 4 to reflect this (lines 356 – 358 in revised manuscript).

Finally, I agree with RC1, who writes " From you figures it shows that many of your images have already been geolocated – this information is not transferred to the meta-data. I recommend that this location data is added to the metadata, or alternatively a supplementary file with image coordinates is attached to the manuscript.

From the response to RC1: The locations used in the maps and based on the image descriptions and images themselves are approximate locations – exact locations would require further ground trothing. Some images allow fairly precise determination of location while others only allow for a general location. With those caveats, we have included coordinates for major features in the images in the supplemental metadata table. Further discussion of how we chose which coordinates to use can be found in lines 270 – 277 in the revised manuscript.

Technical corrections
28: correct typo and formatting in Hill et al.

Done (line 28 in revised manuscript).

102: think it needs a paragraph break after this line.

Done.

108: missing a close paren.

Done.

221: typo "originally size"

Corrected to "original size".

**Response to RC #1**

We would like to thank Anders Anker Bjørk for his time and thoughtful review of the manuscript.  In our response, the reviewer comments are shown in black while our answers are shown in blue.  We also reference the relevant lines in the revised manuscript, in which changes are highlighted.

1) It is not clear to me why the images need to be in two different online archives – each with different level of metadata and (apparently also) different file sizes. I will recommend that you only link to the archive with the most metadata and the largest file sizes/original scans.

We now link to the Cornell library archive, which will be the archive of record.  We have removed discussion of the ARTSTOR archive in the manuscript to avoid confusion.  As noted below, the Cornell archive now have the full sized TIFF images as the default download option, while the ARTSTOR archive has only the downsampled jpg files.

2) I was not able to find the original scan sizes in ARTSTOR, here the images were also c. 1.5 mb as in the Cornell Library. For a 4x5 inch glass plate scanned at high resolution this is very downsized. It would be great with a link or DOI for the high resolution images.

The full resolution TIFF images are now available as the default download through the Cornell archive.  The average size of the TIFF images is 135 mb.

3) From you figures it shows that many of your images have already been geolocated – this information is not transferred to the metadata. I recommend that this location data is added to the metadata, or alternatively a supplementary file with image coordinates is attached to the manuscript.

The locations used in the maps and based on the image descriptions and images themselves are approximate locations – exact locations would require further ground trothing.  Some images allow fairly precise determination of location while others only allow for a general location.  With those caveats, we have included coordinates for major features in the images in the supplemental metadata table.  Further discussion of how we chose which coordinates to use can be found in lines 270 – 277 in the revised manuscript.

4) In your manuscript you call it the R.S. Tarr Collection – in the online archives it is called "Historic Glacial Images of Alaska and Greenland". I find this title misleading as there are images from other locations and other subjects than glaciers too.

We agree that this is not the most descriptive title – it was chosen before the full extent of the digitized images was known.  We have begun a conversation with the Cornell archives about modifying the name, but any changes will be in the longer term.  Any modification of the name will not impact the access link to the collection.

5) The historical images in your figures do not appear to do the material justice. I would like to see a better processing of the photos with more contrast, also the cropping and set-up of the photos can be improved.

Processing the images proved to be a challenge as a number of the originals were degraded after being neglected for a century. The archive staff in charge of digitization tried a number of strategies, including scanning at multiple exposures. We have added additional discussion of this in the manuscript (lines 216 – 222). The master layered images could be reprocessed. We plan to pursue this option, but we will have to secure additional funding before the processing can be performed.

6) Lastly, I am looking more for information on the completeness of the Tarr collection. Are there other images/material from the expeditions in other archives?

We have expanded our discussion in Section 4 (lines 300 – 319) about complimentary collections. Previously, we had mentioned materials from the expeditions housed at the University of Alaska and the National Snow and Ice Data Center. We have now found materials from the expeditions at the University of Colorado, the Anchorage Museum at Rasmuson Center, and the National Geographic Image Collection.

[revised manuscript text omitted]

---

## Author Comment (AC2) · 4 Nov 2019

The comment was uploaded in the form of a supplement:
https://www.earth-syst-sci-data-discuss.net/essd-2019-44/essd-2019-44-AC2-
supplement.zip